# Feasibility of an 8-Week Home-Based Sensory Perception Training Game for People with Fibromyalgia: A Pilot Study

**DOI:** 10.3390/s25010134

**Published:** 2024-12-29

**Authors:** Christophe Demoulin, Chloé Costes, Mélanie Sadok, Stéphanie Grosdent, Jean-François Kaux, Marc Vanderthommen

**Affiliations:** 1Department of Physical Activity and Rehabilitation Sciences, University of Liege, 4000 Liege, Belgium; 2Department of Physical Medicine and Rehabilitation, University Hospital Centre, 4000 Liege, Belgium; 3Faculty of Motor Sciences, Université Catholique de Louvain-La-Neuve, 1348 Louvain-la-Neuve, Belgium

**Keywords:** fibromyalgia, pain, somatosensory training, tactile discrimination

## Abstract

People with fibromyalgia syndrome (FMS) may have difficulty attending rehabilitation sessions. We investigated the feasibility (adherence and satisfaction) of implementing an 8-week home-based somatosensory, entirely remote, self-training programme using the TrainPain smartphone app in people with FMS. The secondary aim was to evaluate the effect on pain symptoms. The training was performed 15 min/day, 7 days/week for 8 weeks. Participants identified the number of vibrations emitted by vibrotactile pods positioned on the most painful site and the contralateral side of the body. They completed the Brief Pain Inventory before, during (4 weeks), and at the end of the 8-week programme. At 8 weeks, they also rated satisfaction and the overall perceived change. The app recorded session completion. Of the 34 individuals recruited, 29 (mean, age 46 [SD] 9 years; 27 women; median duration of symptoms 7 [5;10] years) completed all assessments. Participants completed 75% of sessions and rated the programme easy-to-use and enjoyable, 94% would recommend the programme, and 38% reported a very strong improvement at 8 weeks. Pain intensity reduced from pre to post (effect size 0.77), as did interference (effect size 0.7 to 1.17). This treatment could be a useful addition to a multidisciplinary, multicomponent approach to FMS.

## 1. Introduction

Fibromyalgia syndrome (FMS) is a chronic illness that affects between 0.4% and 9.3% of the global population [1]. The main symptom of the syndrome is widespread pain. The pain of patients with FMS is thought to result mainly from neuro-immune dysfunctions, including structural and functional changes in the central nervous system (CNS) which disrupt pain mechanisms and is classified as nociplastic pain [2]. The changes in the CNS include a disturbance in the nociceptive facilitatory pathways, a dysfunction of the inhibitory pathways that regulate nociception, and an overactivation of sensory processing in the CNS [3,4].

FMS is also characterised by other symptoms, including hypersensitivity to touch, severe fatigue, sleep disorders, and sometimes cognitive impairments and concentration problems, which may lead to depression and anxiety [5]. These multiple symptoms have a considerable, negative impact on the quality of life of affected individuals [5].

Nociplastic conditions like fibromyalgia are characterised by changes in the somatosensory cortex and impairments in various aspects of somatosensory function [6,7,8]. These brain changes are correlated with pain outcomes [9]. Evidence suggests that tactile acuity training could reduce chronic pain [10] as well as improve function [11]. A systematic review of 10 randomised controlled trials involving different methods of somatosensory training [12] concluded that it appears to be effective, but conclusions are limited by considerable heterogeneity between studies and the need for higher-quality studies.

The TrainPain technology platform was developed to support independent, home-based training of somatosensory function within a game-like setting. The game is played using a connected device with 2 vibrotactile pods that are operated via Bluetooth by a smartphone application. A recent pilot study found that the system was feasible for use by people with FMS and that it had a moderate effect on pain [13]. However, that pilot study only included 1 month of training and suggested the need for longer training periods to achieve significant results. Additionally, in that study, the first session, involving an explanation of the treatment, familiarisation and positioning of the vibrotactile pods, was performed in the presence of a therapist; we wished to assess the feasibility of performing the whole training programme remotely, with no physical contact with a therapist, over 8 weeks. If successful, this remote sensory training could help overcome the challenges individuals with pain face in attending in-person sensory rehabilitation appointments.

The main objective of this pilot study was to evaluate the feasibility (adherence and satisfaction) of an 8-week somatosensory, entirely remote, self-training programme using the TrainPain smartphone application in people with FMS. The secondary aim was to perform a preliminary evaluation of the effect on pain and its interference. We hypothesised that participants would adhere to the programme and that it would be associated with a good level of satisfaction. We further hypothesised that pain intensity and interference (measured using the Brief Pain Inventory) would be reduced after completion of the programme and that participants would perceive an overall positive change in their condition.

## 2. Materials and Methods

### 2.1. Design

We conducted a pre–post observational study of an 8-week somatosensory training programme. The study was conducted by Liege University, Belgium, and data were collected between April 2023 and April 2024.

The equipment required for the study was loaned to the participant for the study duration. Once the training was completed, they were asked to return the equipment, and they received 50 euros for their participation.

All the participants were informed of the objective of the project and consented to their participation. The study was approved by the Liege University Hospital Human Ethics Committee (20 March 2023: B7072023000017).

### 2.2. Participants

French-speaking individuals with FMS were recruited in Belgium and France through advertisements placed in hospitals, healthcare clinics, and the social media accounts of some FMS associations.

Inclusion criteria were having a diagnosis of FMS made by a specialist physician (rheumatologist or physical medicine and rehabilitation physician) according to the 2016 criteria [14], a mean self-reported pain intensity during the previous week of ≥4/10 on a numerical rating scale, aged 18–65 years, and agreeing to install an app on their smartphone for the training.

Exclusion criteria were difficulty understanding instructions (or French) or an intellectual deficit, pregnancy or in the post-partum period, diagnosis of a neurological disease (stroke, epilepsy, or peripheral neuropathy) that could affect the somatosensory system, having participated in the somatosensory rehabilitation programme we previously proposed using this technology, and recent changes to their FMS treatment (within the last 6 months).

### 2.3. Study Schedule

Figure 1 illustrates the study schedule. At the end of an information session (conducted remotely by videoconference to explain the purpose and procedures of the study to the participants) and after verifying that the inclusion and exclusion criteria were met, people wishing to participate in the study were invited to sign the information and consent form. They then provided their contact information to receive the necessary training materials by post. The device (TrainPain Inc., Wilmington, DE, USA) included a box linked by wires to 2 vibrotactile pods (Figure 2) operated via Bluetooth by a smartphone application. An electric cable for charging the box was included, along with adhesive bands to secure the stimulators in place.

An email asking the participant to download the TrainPain app on their smartphone (which connects via Bluetooth to the box and is required for training) and to complete a battery of questionnaires was sent the day before the first training session (Day 1). Participants were also sent a tutorial on how to use the equipment.

### 2.4. Assessments

Assessments were conducted before the start of the training (pre), in the middle of the programme (4 weeks) and at the end of the programme (post). Participants completed the questionnaires at home.

Feasibility was measured by participant adherence to the programme and their satisfaction with the programme. Pain and its interference were measured with the Brief Pain Inventory (BPI), and participants also rated their overall perceived change.

#### 2.4.1. Pre Assessment

The questionnaire battery included a sociodemographic questionnaire (age, gender, employment status, duration of pain), a question aimed at identifying the most painful body region (for appropriate placement of the electrodes) and the BPI.

The BPI is widely used in pain research and clinical trials, including those on chronic pain conditions such as fibromyalgia [15]. The Initiative on Methods, Measurement, and Pain Assessment in Clinical Trials (IMMPACT) group recommends the BPI for inclusion in any clinical trial evaluating pain [16]. In contrast with the Visual Analogue Scale (VAS) and similar unidimensional tools that provide a quick snapshot of pain intensity, the BPI captures the complexity of the pain experience, including severity and interference with daily life and functioning [17,18]. The BPI has good internal consistency, test retest reliability and construct validity, and is responsive to change [19]. We used the French version of the BPI [20]. It assesses pain intensity using 4 numerical pain scales ranging from 0 (no pain) to 10 (maximum pain), including the highest and lowest pain intensities during the previous week, average pain levels, and pain at the time of the assessment. The BPI also contains 7 items exploring the impact of the pain on the person’s daily life (i.e., general activity, walking, work, mood, enjoyment of life, relations with others, and sleep). To score the BPI, the assessor calculates the average of the 4 severity items (pain severity subscale) and the average of the 7 interference items (pain interference subscale).

#### 2.4.2. Intermediate and Post Assessments

The 4-week and post assessments included the same battery of questionnaires as the pre-test (except for the general information). In addition, participants were also asked to rate their perceived change in their general fibromyalgia-related symptoms since the beginning of the study on a 7-point overall perceived change scale. The post evaluation also included a satisfaction questionnaire with 3 questions: “Did you find the game fun to play?” (0 = not at all, 10 = lots of fun), “Was it easy for you to perform the programme daily?” (0 = not at all, 10 = very easy), and “If a friend or family member had a similar pain problem, would you recommend the programme to them?” (Yes/No)?

#### 2.4.3. Adherence

Data regarding sessions performed were recorded by the app.

### 2.5. The TrainPain App

The TrainPain app and device provide a variety of somatosensory training exercises designed to improve sensory attention efficiency, flexibility, and the ability to filter out tactile distractions, helping to counteract body-related attentional biases. Patterns embedded in the stimulations offer clues for solving game-based puzzles (e.g., matching the number of pulses to the target bubble to pop) (Figure 3). To progress, users must accurately discriminate target stimulations and ignore irrelevant sensory input. An adaptive algorithm dynamically adjusts the difficulty, creating a progressively challenging experience tailored to the user’s performance.

The game platform consists of 800 levels, with the training process starting at level 1. Once a level is successfully completed, it is unlocked, enabling participants to progress to the next level. As the levels advance, the sensory exercises are adjusted to become easier or more challenging, depending on the participant’s success (accuracy).

### 2.6. Training

The training period lasted for 8 weeks. Participants were asked to train daily (7 days per week) for up to 15 min per day.

The first session was conducted via videoconference with an investigator. After confirming the app’s setup and functionality and providing instructions on recharging the sensor pods, participants were guided to attach one vibrotactile pod to their most painful area and the other to the corresponding area on the opposite side of the body, using adhesive patches. They were instructed to keep the pods in the same location for each session unless the pain in that area subsided or became too sensitive to touch. In such cases, participants could contact the investigator to reposition the pod to the next most painful area.

If participants could hear the pods vibrating, they were advised to wear headphones and to listen to the game music or their own music to ensure the focus was on the tactile sensations rather than the sound.

After launching the application, the participant underwent a familiarisation session provided as a tutorial within the app. The familiarisation took place during the first session and could only be performed once. However, the tutorial remained accessible, though its use was not usually necessary thereafter. The objective of the tutorial was to teach the participant how the application and game functioned.

Since the tutorial took several minutes to complete, the first session included about 8 min of training tasks instead of the 15 min allocated for subsequent sessions. This initial session ensured that participants could use the training app and understood how to play the game.

Thereafter, participants received a daily reminder from the application at 6 PM to encourage them to play. However, they were free to train at the time they felt most comfortable and focused. A minimum interval of 10 h was required between training sessions (the app was locked during this time). Training sessions were limited to a maximum duration of 15 min per day. However, if a level had been started, participants were allowed to complete it before the app locked.

Participants could play the game in any comfortable position, provided that the sensors were not in contact with a hard surface (for example, not lying on their back or sitting against a backrest if the sensors were positioned on their back). Participants were encouraged to approach each training session as a relaxed and playful experience, focusing on enjoyment rather than exerting intense effort or stress.

### 2.7. Statistical Analysis

Statistical analyses were carried out using Excel and IBM SPSS Statistics version 28.0 software. The normality of the outcomes was checked with a histogram, a QQ-plot, and the Shapiro–Wilk test. Outcomes are expressed as mean (standard deviation) for data that followed a normal distribution (median and interquartile range otherwise).

Repeated Measures ANOVA and post hoc Bonferroni tests were used to analyse the changes in the outcomes after the training programme. Effect size was calculated as ES = (Meanpost − Meanpre)/SDdiff, where SDdiff is the standard deviation of the paired differences. A *p*-value < 0.05 was considered significant for all the analyses.

## 3. Results

### 3.1. Participants

A total of 34 individuals with FMS (94% women) were recruited for this study and completed the pre-test questionnaires. Twenty-nine participants completed the entire study and were included in the analysis; their characteristics are presented in Table 1. Among the 5 dropouts, 1 never began the training due to technical issues, one stopped after only 2–3 days of training for personal reasons, and 3 others left the study during the second month (2 for personal reasons and one because of an unrelated skin condition).

### 3.2. Pain Location

Figure 4 shows the location of the area identified by participants as either the most painful or one of the most painful, which was then used for positioning the vibrotactile pods (the second pod was placed on the contralateral side). The most frequently selected areas were the thighs, the neck and the interscapular region. One participant changed the location after 2 days of training due to discomfort in the area initially chosen.

### 3.3. Adherence

The mean (SD) number of sessions (i.e., days of training) performed over the 8 weeks was 46.5 (15.3); therefore, the adherence rate was 76% (25%). Only five participants performed fewer than five sessions per week.

### 3.4. Satisfaction

The median (IQR) score for the fun aspect of the game was 9 (6;10), range 4–10 and for ease of performing the programme daily was 9 (8;10), range 2–10. Twenty-seven (94%) of the participants said they would recommend the programme to a friend or relative with a similar pain condition.

### 3.5. Overall Perceived Change

Table 2 shows the overall change perceived by participants at the 4-week and post assessments compared to their condition before starting this study. At 4 weeks, around one-quarter of participants considered they were much better and by 8 weeks (post), more than one-third considered they were much or very much better. No participants reported a worsening of their condition.

### 3.6. Pain Intensity

Table 3 shows the change in pain according to the BPI ratings. Pain intensity reduced from pre to post for the most intense pain experienced in the last 24 h and average pain.

### 3.7. Interference of Pain with Daily Life

Table 4 shows the change in the scores of the BPI items that rate the interference of pain with daily life. All the items improved significantly between pre and 4 weeks and pre and post.

## 4. Discussion

This study aimed to evaluate the feasibility and effectiveness of a new, 8-week (15 min per day) somatosensory perceptual training programme performed using a smartphone and vibrotactile pods in people with FMS. The study showed that the programme could be performed entirely remotely at home, with no face-to-face contact and the equipment sent by post. Participants adhered well to the programme and found the game enjoyable and easy to use. Positive effects were found on pain and interference scores, with effect sizes >0.7 for average pain intensity, highest pain intensity in the last 24h, and all of the interference-related items of the BPI. Furthermore, 38% of participants reported a strong or very strong improvement in their condition at the end of the programme.

Most of the participants were females, aged between 40 and 60 years, and had experienced fibromyalgia for more than 5 years. Around 50% were still working, but the other 50% were unable to work or were unemployed due to their pain. These prevalences are concordant with reports in the literature [5,21,22]. Pain locations varied among participants but often involved the thighs and spine (neck, interscapular, and dorsal regions), which corresponds with other reports in the literature [13].

In accordance with our hypotheses that participants would adhere to the programme and that it would be associated with a good level of satisfaction, this study demonstrated the feasibility of an entirely remote, 2-month programme (vs. 4 weeks with a face-to-face programme initiation in the previous pilot study [13]) and that it was appreciated. With 75% of sessions performed by our participants, their adherence was in the range of the moderate to high attendance (73% to 87.20%) of supervised exercise sessions reported in people with FMS in a literature review [23]. This result is very encouraging for an entirely remote exercise programme. The high rate found in our study corresponds with the high level of satisfaction with the programme (median 9/10). These results could potentially be explained by the fact the participants found the training enjoyable and easy to complete, and that 80% perceived an overall positive change in their condition. Furthermore, 94% stated they would recommend the programme to others with similar pain. These results confirm those in the previous study [13].

With regard to the clinical effects, we hypothesised that pain intensity and interference would be reduced after completion of the programme and that participants would perceive an overall positive change in their condition. In line with our expectations, our study showed significant improvements in most pain and interference scores between the pre-test and post-test, characterised by a large (>0.8), or nearly large (0.7 < effect size < 0.8) effect size for most outcomes. Furthermore, the moderate effect size for pain intensity was larger than those reported in some systematic reviews for pharmacological approaches to FMS [24].

Although the improvement primarily occurred during the first 4 weeks, consistent with the results of the previous pilot study [13], an additional 4 weeks of training was associated with the maintenance or further (sometimes significant) improvement of the variables. A follow-up of participants after this 2-month programme would be necessary to verify if the effect was maintained after the training stopped. A deterioration was observed in some individuals 1 month after the end of the 4-week training period in the pilot cross-over study [13].

These effects of the training programme are confirmed by the analysis of the overall perceived change, where 42%, 28%, and 10% of participants reported slight, strong, and very strong improvements, respectively, after the 8-week training. These results are even more encouraging than those observed in the previous pilot study, in which 28% of subjects reported a slight improvement and 17% strong improvement at 4 weeks [13]. Impressive effects were observed in some participants, including one who reported experiencing 0/10 pain for the first time since her diagnosis of FMS after trying many previous treatments.

Although the results of this study, which will need to be confirmed in a randomised controlled trial with a larger sample, suggest a real clinical effect of this training, they do not imply that this approach is beneficial for all individuals with FMS or that it can be used as a standalone treatment. Firstly, the management of FMS must be multidisciplinary [25,26]. FMS is a chronic pain syndrome that is often associated with psychosocial factors that can contribute to the persistence of pain [27] and physical deconditioning. Somatosensory perceptual training should ideally be combined with other non-invasive approaches (selected according to the individual’s needs and preferences) that have also shown some efficacy, such as pain neurophysiology education [28], graded physical activities [29,30], relaxation techniques [31], and cognitive-behavioural therapy [32] (to manage potential stress, anxiety, depression, etc.). A multimodal, and even multidisciplinary treatment approach combining individualised care with physical and psychosocial interventions is therefore recommended. The TrainPain system could be provided to individuals with FMS as part of such a programme, with the aim of contributing to reducing pain symptoms. Remote use of the app could help individuals take charge of at least a part of their own rehabilitation and improve their level of self-efficacy.

This study has several limitations, such as a small sample size, the absence of a control group, and the lack of follow-up after the training period. We did not monitor whether the participants adhered to the guidance to use headphones if they could hear the pod vibrations. The correlations between changes in somatosensory discrimination capacity induced by the training and clinical improvements will need to be studied further. This could be performed using the test that was recently developed using the same device and vibrotactile pods [8].

## 5. Conclusions

This study demonstrated the feasibility of an entirely remote, 8-week somatosensory self-training programme using the TrainPain smartphone application. Participant satisfaction was high, and the preliminary results for the effectiveness on pain and its interference were positive. This somatosensory perceptual training method appears to be feasible remotely and over 8 weeks, and it seems to result in clinically significant improvements in pain and interference scores. This treatment could be a useful addition to a multidisciplinary, multicomponent approach to the management of FMS.

## Figures and Tables

**Figure 1 sensors-25-00134-f001:**
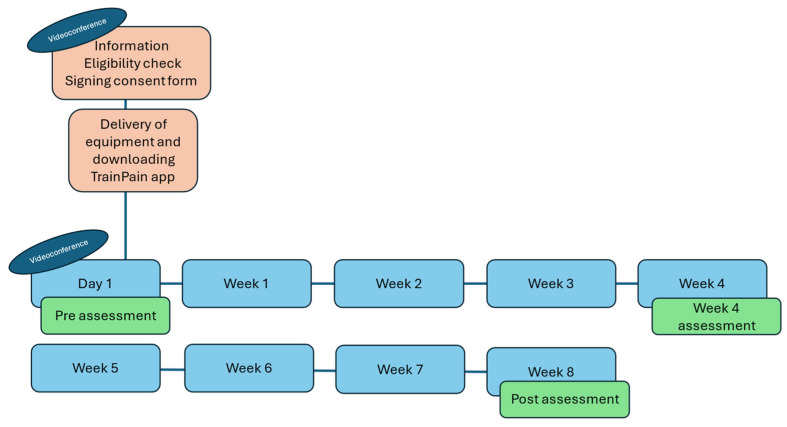
Study schedule.

**Figure 2 sensors-25-00134-f002:**
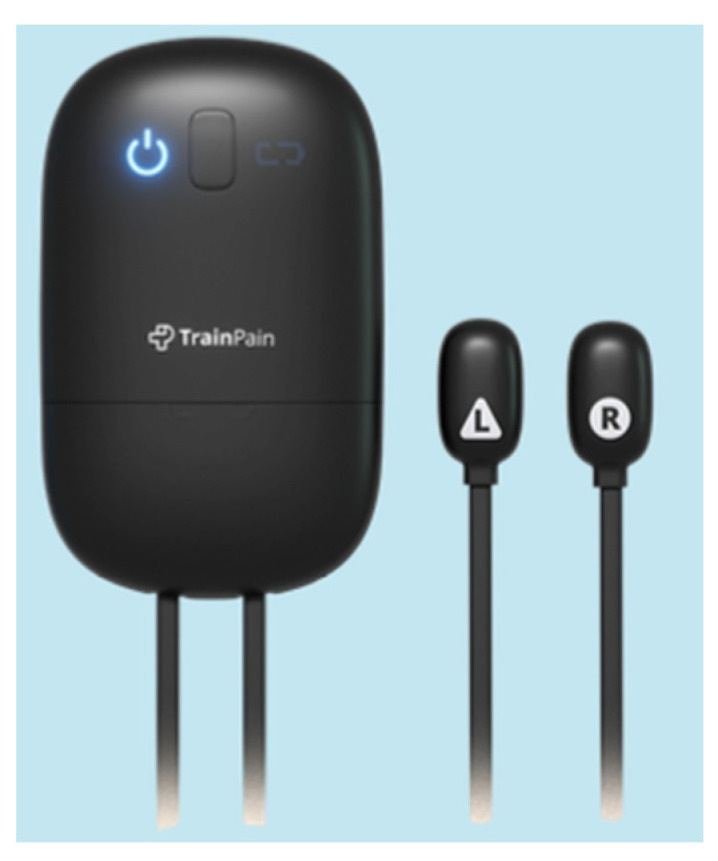
The TrainPain device—box and 2 vibrotactile pods.

**Figure 3 sensors-25-00134-f003:**
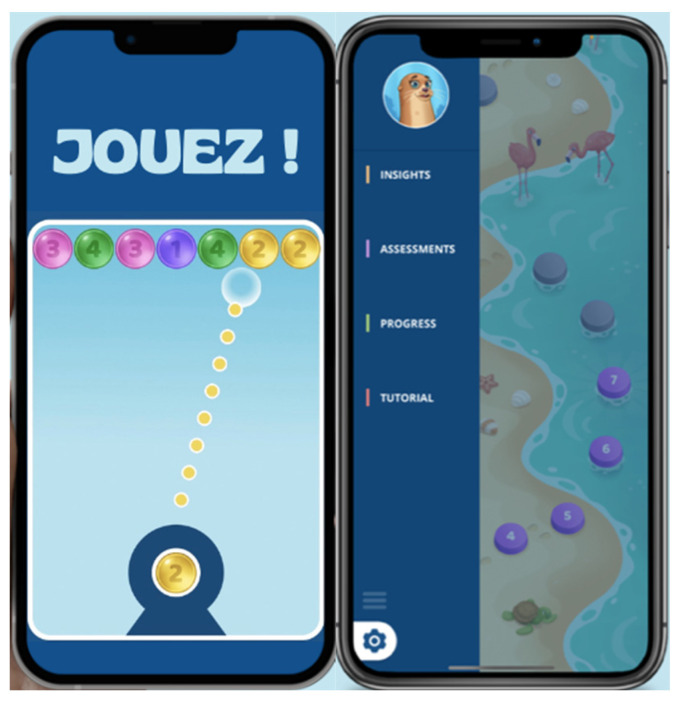
The TrainPain app.

**Figure 4 sensors-25-00134-f004:**
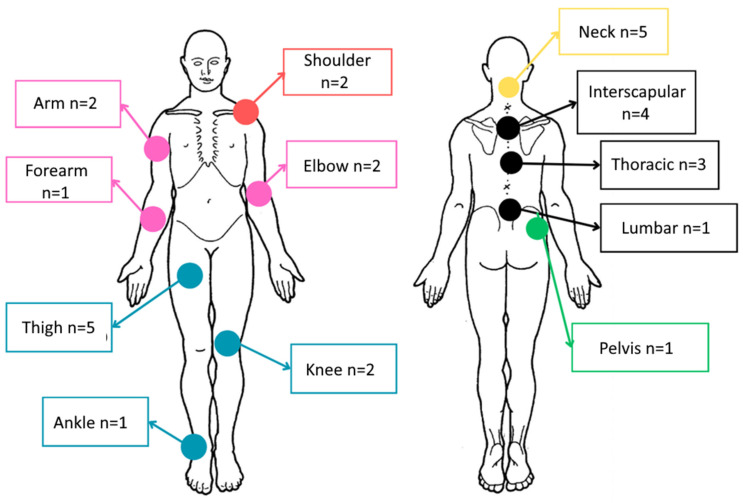
Body-chart showing the locations of pain chosen by the 29 participants for positioning the vibrotactile pods.

**Table 1 sensors-25-00134-t001:** Participant characteristics (N = 29).

Characteristic		
Age, years, mean (SD) [range]		46 (9) [25–62]
Gender	Male	2 (7%)
	Female	27 (93%)
Professional situation	Working	16 (55%)
	Unemployed	4 (14%)
	On sick leave	9 (31%)
Duration of symptoms, years, median (IQR) [range]		7 (5;10) [1–25]

**Table 2 sensors-25-00134-t002:** Self-rating of overall perceived change (N = 29).

	4 Weeks	Post
Extremely better	0% (n = 0)	10% (n = 3)
Much better	24% (n = 7)	28% (n = 8)
Slightly better	38% (n = 11)	42% (n = 12)
No change	38% (n = 11)	20% (n = 6)
Slightly worse	0% (n = 0)	0% (n = 0)
Extremely worse	0% (n = 0)	0% (n = 0)

**Table 3 sensors-25-00134-t003:** Brief Pain Inventory pain scores at each time point.

Pain Intensity	Pre (N = 29)	4 Weeks (N = 29)	Post (N = 29)	*p*-Value	Effect Size (Pre-Post)	Delta Pre-Post
Highest in previous 24 h	7 (7;8) [4–10]	5 (4;7) [0–9]	5(3;7) [1–10]	<0.001 (Pre > 4-w, Pre > Post)	0.73	2.34
Lowest in previous 24 h	3 (2;4) [0–7]	2(2;3) [0–8]	3(1;4) [0–8]	0.375	0.18	0.45
Now	5 (2;7) [0–9]	4(2;6) [0–8]	4(2;5) [0–10]	0.216	0.12	0.38
Average pain	5.86 (1.33) [2–8]	5.41 (1.74) [1–8]	4.45 (2.05) [0–8]	0.008 (Pre > Post)	0.77	1.41

Data are median (IQR) [range] except for average pain which are mean (SD) [range] and Delta Pre–Post data which are mean (SD).

**Table 4 sensors-25-00134-t004:** Change in the scores of the Brief Pain Inventory (BPI) items that rate the interference of pain with daily life.

BPI Item	Pre (N = 29)	4 Weeks (N = 29)	Post (N = 29)	*p*-Value	Effect Size	Delta Pre-Post *
General activity, median (IQR) [range]	7 (5;8) [2–10]	5 (4;6) [0–8]	4 (2;6) [1–10]	<0.001 (Pre > 4-w, Pre > Post)	0.94	2.38 (2.54)
Normal work, median (IQR) [range]	8(5;9) [1–10]	5(3;7) [0–8]	5(2;7) [1–10]	0.001 (Pre > 4-w, Pre > Post)	0.88	2.21 (2.50)
Enjoyment of life, median (IQR) [range]	7(5;8) [0–10]	5(3;7) [0–9]	4(1;7) [0–10]	0.004 (Pre > 4-w, Pre > Post)	0.96	2.48 (2.6)
Mood, median (IQR) [range]	6(4;8) [0–10]	4(2;6) [0–8]	3(1;5) [0–9]	0.001 (Pre > Post)	1.17	2.55 (2.18)
Walking ability, mean (SD) [range]	5.76 (2.36) [0–9]	4.21 (2.29) [0–8]	3.69 (2.95) [0–10]	0.008 (Pre > 4-w)	0.75	2.07 (2.77)
Sleep, mean (SD) [range]	7.21 (2.74) [0–10]	5.45 (3.02) [0–10]	4.93 (3) [0–10]	0.01 (Pre > 4-w)	0.75	2.28 (3.03)
Relations with other people, mean (SD) [range]	5.31 (2.82) [0–10]	3.83 (2.66) [0–8]	3.59 (3.10) [0–10]	0.050	0.70	1.72 (2.45)

* Delta Pre–Post data are mean (SD).

## Data Availability

The dataset is available on request from the authors.

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
