# Peer review of "Feasibility of an 8-Week Home-Based Sensory Perception Training Game for People with Fibromyalgia: A Pilot Study"

_sensors, 2024, doi:10.3390/s25010134_

Round 1
Reviewer 1 Report
Comments and Suggestions for Authors
This paper describes a study evaluating the performance of an off-the-shelf vibrotactile sensory perception "serious game" under non-laboratory settings. The investigators evaluated both the overall experience of users, and pain reduction effects using self-reported assessments.
Overall, the study design and analysis are straightforward with encouraging results. The authors acknowledge that the results do not provide strong conclusive evidence because it is not a randomized controlled trial.
Based on my review of their cited literature there is a small amount of evidence that somatosensory training is effective in reducing pain, so the findings in this study are not new but represent a contribution to the body of literature.
Here are some aspects of the paper that should be addressed:
- There should be more discussion on why the BPI was used. The BPI seems to have been design to study cancer pain. In the review on SDT that was cited, most studies used the Visual Analog Scale, while the study on fibromyalgia used the Self Assessment of Pain. The BPI seems more comprehensive, but even then there should be a clear justification of the advantages of this scale over the other scales. The advantages should outweigh the cost of not using a scale that allows comparison to prior literature.
- In your introduction, please state the hypotheses that were tested in the study. Clearly relate them to the primary and secondary objectives of your study.
- Regarding the primary objective of assessing feasibility, it is unclear what metrics were used to judge feasibility. The reader can infer them from the subsequent text, but it would be better to state them clearly.
- Regarding the adherence to the protocol: Were there any steps taken to measure the adherence to the protocol? Could these be monitored from in-app data? For example, the guidance to use headphones was optional. How do we know participants adhered to the guidance? For out-of-lab studies, more data points are needed to mitigate the "noisy" data, so I deem such methodological considerations more important given that the sample size of the study is low.
Author Response
|
Response to Reviewer 1 Comments
|
|
We thank the reviewers for taking the time to review our manuscript. Please find a point-by-point response below and the corresponding revisions highlighted in the re-submitted files.
|
|
Point-by-point response to Comments and Suggestions from Reviewer 1
|
|
Comments 1: There should be more discussion on why the BPI was used. The BPI seems to have been design to study cancer pain. In the review on SDT that was cited, most studies used the Visual Analog Scale, while the study on fibromyalgia used the Self Assessment of Pain. The BPI seems more comprehensive, but even then there should be a clear justification of the advantages of this scale over the other scales. The advantages should outweigh the cost of not using a scale that allows comparison to prior literature.
|
|
Response 1: Thank you for pointing this out that we did not sufficiently justify the use of the BPI. The BPI is widely utilized in pain research and clinical trials, including those on chronic pain conditions such as fibromyalgia, as documented in its user guide and several validation studies • The Initiative on Methods, Measurement, and Pain Assessment in Clinical Trials (IMMPACT) group recommends the BPI for inclusion in any clinical trial evaluating pain, further underscoring its validity and relevance in diverse pain populations. While the Visual Analog Scale (VAS) and similar unidimensional tools can provide a quick snapshot of pain intensity, they do not capture the complexity of the pain experience. This is particularly important in conditions like fibromyalgia, where pain’s impact extends beyond intensity to interfere with daily life and functioning. The BPI’s multidimensional approach assesses not only pain intensity (via four Numeric Rating Scale [NRS] items: current, worst, least, and average pain) but also the impact of pain on various aspects of life, including mood, sleep, and daily activities. This makes it a more comprehensive tool, especially for chronic pain intervention studies. • Moreover, research has highlighted challenges patients face in understanding and using the VAS, particularly in chronic pain settings, where scales like the BPI are better suited to capturing the nuanced pain experience. By using the BPI, which includes four NRS pain assessments, we ensure both a thorough pain evaluation and alignment with modern standards in chronic pain research. • Additionally, the U.S. FDA recognizes the BPI as a recommended outcome measure, further validating its appropriateness in this context. The BPI has been validated for use in fibromyalgia and is explicitly discussed in the BPI user guide by its developers as a suitable tool for assessing pain in this population. Its inclusion in this study ensures both clinical relevance and a robust assessment of pain and its impact on quality of life.
We added the following text in the Methods section and added the references cited below:
Line 141: The BPI is widely used in pain research and clinical trials, including those on chronic pain conditions such as fibromyalgia [15]. The Initiative on Methods, Measurement, and Pain Assessment in Clinical Trials (IMMPACT) group recommends the BPI for inclusion in any clinical trial evaluating pain [16]. In contrast with the Visual Analog Scale (VAS) and similar unidimensional tools that provide a quick snapshot of pain intensity, the BPI captures the complexity of the pain experience, including severity and interference with daily life and functioning [17],[18]. The BPI has good internal consistency, test retest reliability and construct validity, and is responsive to change [19]. We used the French version of the BPI [20]. It assesses pain intensity using 4 numerical pain scales ranging from 0 (no pain) to 10 (maximum pain), including the highest and lowest pain intensities during the previous week, average pain levels, and pain at the time of the assessment. The BPI also contains 7 items exploring the impact of the pain on the person’s daily life (i.e., general activity, walking, work, mood, enjoyment of life, relations with others, and sleep). To score the BPI, the assessor calculates the average of the 4 severity items (pain severity subscale) and the average of the 7 interference items (pain interference subscale). [15] Wang, F.; Ruberg, S. J.; Gaynor, P. J.; Heinloth, A. N.; Arnold, L. M. Early Improvement in Pain Predicts Pain Response at Endpoint in Patients with Fibromyalgia. J. Pain 2011, 12 (10), 1088–1094. https://doi.org/10.1016/j.jpain.2011.05.002. [16] Dworkin, R. H.; Turk, D. C.; Farrar, J. T.; Haythornthwaite, J. A.; Jensen, M. P.; Katz, N. P.; Kerns, R. D.; Stucki, G.; Allen, R. R.; Bellamy, N.; Carr, D. B.; Chandler, J.; Cowan, P.; Dionne, R.; Galer, B. S.; Hertz, S.; Jadad, A. R.; Kramer, L. D.; Manning, D. C.; Martin, S.; McCormick, C. G.; McDermott, M. P.; McGrath, P.; Quessy, S.; Rappaport, B. A.; Robbins, W.; Robinson, J. P.; Rothman, M.; Royal, M. A.; Simon, L.; Stauffer, J. W.; Stein, W.; Tollett, J.; Wernicke, J.; Witter, J. Core Outcome Measures for Chronic Pain Clinical Trials: IMMPACT Recommendations. Pain. 2005, pp 9–19. https://doi.org/10.1016/j.pain.2004.09.012. [19] Williams, D. A.; Arnold, L. M. Measures Applied to the Assessment of Fibromyalgia. Arthritis Care Res. 2011, 63 (2), 1487–1495. https://doi.org/10.1002/acr.20531.Measures.
We also realised that we described the outcomes of ‘pain’ and ‘function’ whereas the BPI does not strictly measure function but ‘pain interference’. Therefore, we have changed ‘function’ to ‘interference with function’ throughout the text.
|
|
Comments 2: In your introduction, please state the hypotheses that were tested in the study. Clearly relate them to the primary and secondary objectives of your study.
|
|
Response 2: Thank you for this suggestion. We have added hypotheses. Line 69: The main objective of this pilot study was to evaluate the feasibility (adherence and satisfaction) of an 8-week somatosensory, entirely remote, self-training programme using the TrainPain smartphone application in people with FMS. The secondary aim was to perform a preliminary evaluation of the effect on pain and its interference on function. We hypothesised that participants would adhere to the programme, and that it would be associated with a good level of satisfaction. We further hypothesised that pain intensity and interference (measured using the Brief Pain Inventory) would be reduced after completion of the programme and that participants would perceive an overall positive change in their condition.
|
|
Comments 3: Regarding the primary objective of assessing feasibility, it is unclear what metrics were used to judge feasibility. The reader can infer them from the subsequent text, but it would be better to state them clearly.
|
|
Response 3: Thank you for this important point. We considered the fact that most participants completed the study and were satisfied as indicators of feasibility. However, this was not clearly stated and we did not include adherence data which could also be considered as a measure of feasibility. We have now added adherence data (which was recorded by the app) and clearly stated the indicators of feasibility in the aims and methods.
Line 69: The main objective of this pilot study was to evaluate the feasibility (adherence and satisfaction) Line 129: 2.4 Assessments Assessments were conducted before the start of the training (Pre), in the middle of the programme (4 weeks) and at the end of the programme (Post). Participants completed the questionnaires at home. Feasibility was measured by participant adherence to the programme and their satisfaction with the programme. Pain and its interference were measured with the Brief Pain Inventory (BPI), and participants also rated their overall perceived change.
Line 255: 3.3 Adherence The mean (SD) number of sessions (i.e., days of training) performed over the 8 weeks was 46.5 (15.3), therefore adherence rate was 76% (25%). Only 5 participants performed fewer than 5 sessions per week.
|
|
Comments 4: Regarding the adherence to the protocol: Were there any steps taken to measure the adherence to the protocol? Could these be monitored from in-app data? For example, the guidance to use headphones was optional. How do we know participants adhered to the guidance? For out-of-lab studies, more data points are needed to mitigate the "noisy" data, so I deem such methodological considerations more important given that the sample size of the study is low. |
|
Response 4: Thank you for this point which led us to include adherence data as an indicator of feasibility. Use of the app was indeed monitored by the app. Please see Response 3 for details. With regards to whether participants followed the guidance about the use of headphones, we did not monitor this. We have added a sentence about this in the limitations section.
Line 362: We did not monitor whether the participants adhered to the guidance to use headphones if they could hear the pod vibrations.
|
|
|
Reviewer 2 Report
Comments and Suggestions for Authors
Thank you very much for giving me the opportunity to review this interesting paper entitled Feasibility of an 8-week Home-Based Sensory Perception Training Game for People with Fibromyalgia: A Pilot Study. Altogether it is an interesting and well written study that aimed to investigate the feasibility of an 8-week home-based somatosensory, entirely remote, self-training programme to reduce pain in people with fibromyalgia syndrome (FMS) as well as to perform a preliminary evaluation of its effect on pain and function among 29 individuals (of a total of 34 individuals recruited) with FMS. Repeated Measures ANOVA and post hoc Bonferroni tests were conducted to analyze the changes in the aforementioned outcomes after the training programme. The results obtained showed a reduction in pain intensity and an improvement in function from pre to post assessment. As authors noted, this 8-week home-based somatosensory, entirely remote, self-training programme could be a useful addition to a multidisciplinary, multicomponent approach to FMS.
The objective is in general well defined, and citations are in general current.
The description of the app and the training program are accurate, and the statistical analyses are coherent.
Say this, here are a few comments:
1. You could place the question addressed in a broad context in their abstract, what makes it so relevant in the context of fibromyalgia to study a remote training programme? Perhaps it could be useful if you review the wording of the abstract, reconsidering also the relevance or not of providing in it some of the descriptive data it contains.
2. The introduction is well written, however in my humble opinion it would be interesting to reinforce it, for example offering in greater detail the results of the studies they mention, as well as what is expressed in lines 45-46. It would also be interesting to describe in more detail already in the introduction the results of the pilot study referred to in lines 50-56, as they seem to do later in lines 261-263.
3. In my humble opinion, it would be important to rethink the titles of the sections and their structure in the materials and methods section to make the reading easier for the potential reader, highlighting the value of the work conducted. For example, the expression procedure instead of design could be used and within this section offer everything that has to do with it.
4. I am more used to the fact that even minimally in the participants section (lines 79-83) some data will be offered describing them, (although I understand that this broader description is made in the results section).
5. It would be advisable to describe the measures used in more detail, for example by offering some examples of the items they include as well as reliability data.
6. First, in the results section you could provide a short introductory paragraph to help the reader. In addition, you could restructure the way in which the results are presented (e.g., by type of analysis) and their order (you offer satisfaction results that you only measure at the end of the program before those of the change between pre and post).
7. The discussion section might be improved, first, remembering what the objective of their work is.
8. The conclusions section might be improved, for example offering in more detail what are the potential practical implications of the results obtained.
9. Please check the references both in the text and references section, it does not seem to completely follow the format indicated by the journal. Likewise, if possible, it could be interesting to incorporate a greater number of references, especially more recent ones from 2024.
Author Response
|
Response to Reviewer 2 Comments
|
|
We thank the reviewers for taking the time to review our manuscript. Please find a point-by-point response below and the corresponding revisions highlighted in the re-submitted files.
|
|
Point-by-point response to Comments and Suggestions from Reviewer 2
|
|
Comments 1: You could place the question addressed in a broad context in their abstract, what makes it so relevant in the context of fibromyalgia to study a remote training programme? Perhaps it could be useful if you review the wording of the abstract, reconsidering also the relevance or not of providing in it some of the descriptive data it contains. |
|
Response 1: Thank you for this suggestion. We have included a sentence about why remote training could be useful for people with fibromyalgia in the opening of the abstract. Line 18: People with fibromyalgia syndrome (FMS) may have difficulty attending rehabilitation sessions. We also rewrote parts of the abstract to improve clarity. However, we did not remove the descriptive data because we believe these data are important for interpretation of the generalisability of the study. Line 18: People with fibromyalgia syndrome (FMS) may have difficulty attending rehabilitation sessions. We investigated the feasibility (adherence and satisfaction) of implementing an 8-week home-based somatosensory, entirely remote, self-training programme using the TrainPain smartphone app in people with FMS. The secondary aim was to evaluate the effect on pain symptoms. The training was performed 15 minutes/day, 7 days/week for 8 weeks. Participants identified the number of vibrations emitted by vibrotactile pods positioned on the most painful site and the contralateral side of the body. They completed the Brief Pain Inventory before, during (4 weeks) and at the end of the 8-week programme. At 8 weeks, they also rated satisfaction and the overall perceived change. The app recorded session completion. Of the 34 individuals recruited, 29 (mean, age 46 [SD] 9 years; 27 women; median duration of symptoms 7 [5;10] years) completed all assessments. Participants completed 75% of sessions and rated the programme easy-to-use and enjoyable; 94% would recommend the programme and 38% reported a very strong improvement at 8 weeks. Pain intensity reduced from Pre to Post (effect size 0.77), and function improved (effect size 0.7 to 1.17). This treatment could be a useful addition to a multidisciplinary, multicomponent approach to FMS.
|
|
Comments 2: The introduction is well written, however in my humble opinion it would be interesting to reinforce it, for example offering in greater detail the results of the studies they mention, as well as what is expressed in lines 45-46. It would also be interesting to describe in more detail already in the introduction the results of the pilot study referred to in lines 50-56, as they seem to do later in lines 261-263.
|
|
Response 2: We thank you for these suggestions. We have added some details of some trials and 2 new references.
Line 48: Nociplastic conditions like fibromyalgia are characterized by changes in the somatosensory cortex and impairments in various aspects of somatosensory function [6]–[8]. These brain changes are correlated with pain outcomes [9]. Evidence suggests that tactile acuity training could reduce chronic pain [10] as well as improve function [11]. A systematic review of 10 randomised controlled trials involving different methods of somatosensory training [12] concluded that it appears to be effective, but conclusions are limited by considerable heterogeneity between studies and the need for higher-quality studies. [10] Barker, K. L.; Elliott, C. J.; Sackley, C. M.; Fairbank, J. C. T. Treatment of Chronic Back Pain by Sensory Discrimination Training. A Phase I RCT of a Novel Device (FairMed) vs. TENS. BMC Musculoskelet. Disord. 2008, 9, 1–8. https://doi.org/10.1186/1471-2474-9-97. [11] Kälin, S.; Rausch-Osthoff, A. K.; Bauer, C. M. What Is the Effect of Sensory Discrimination Training on Chronic Low Back Pain? A Systematic Review. BMC Musculoskelet. Disord. 2016, 17 (1), 1–9. https://doi.org/10.1186/s12891-016-0997-8.
The results of the pilot study are already mentioned in the introduction
Line 58: A recent pilot study found that the system was feasible for use by people with FMS and that it had a moderate effect on pain [13].
|
|
Comments 3: In my humble opinion, it would be important to rethink the titles of the sections and their structure in the materials and methods section to make the reading easier for the potential reader, highlighting the value of the work conducted. For example, the expression procedure instead of design could be used and within this section offer everything that has to do with it. |
|
Response 3: We understand your point of view, however we believe that the titles and sections that we have used in the Methods section help the reader to find information about the different aspects of the study. |
|
Comments 4: I am more used to the fact that even minimally in the participants section (lines 79-83) some data will be offered describing them, (although I understand that this broader description is made in the results section). |
|
Response 4: Many journals prefer participant data to be placed in the results section because at the time the study was performed and the methods were defined, these data were unknown. Therefore, we have left these data in the results section.
|
|
Comments 5: It would be advisable to describe the measures used in more detail, for example by offering some examples of the items they include as well as reliability data. |
|
Response 5: We thank you for pointing this out. We have added information about the Brief Pain Invertory.
Line 141: The BPI is widely used in pain research and clinical trials, including those on chronic pain conditions such as fibromyalgia [15]. The Initiative on Methods, Measurement, and Pain Assessment in Clinical Trials (IMMPACT) group recommends the BPI for inclusion in any clinical trial evaluating pain [16]. In contrast with the Visual Analog Scale (VAS) and similar unidimensional tools that provide a quick snapshot of pain intensity, the BPI captures the complexity of the pain experience, including severity and interference with daily life and functioning [17],[18]. The BPI has good internal consistency, test retest reliability and construct validity, and is responsive to change [19]. We used the French version of the BPI [20]. It assesses pain intensity using 4 numerical pain scales ranging from 0 (no pain) to 10 (maximum pain), including the highest and lowest pain intensities during the previous week, average pain levels, and pain at the time of the assessment. The BPI also contains 7 items exploring the impact of the pain on the person’s daily life (i.e., general activity, walking, work, mood, enjoyment of life, relations with others, and sleep). To score the BPI, the assessor calculates the average of the 4 severity items (pain severity subscale) and the average of the 7 interference items (pain interference subscale). [15] Wang, F.; Ruberg, S. J.; Gaynor, P. J.; Heinloth, A. N.; Arnold, L. M. Early Improvement in Pain Predicts Pain Response at Endpoint in Patients with Fibromyalgia. J. Pain 2011, 12 (10), 1088–1094. https://doi.org/10.1016/j.jpain.2011.05.002. [16] Dworkin, R. H.; Turk, D. C.; Farrar, J. T.; Haythornthwaite, J. A.; Jensen, M. P.; Katz, N. P.; Kerns, R. D.; Stucki, G.; Allen, R. R.; Bellamy, N.; Carr, D. B.; Chandler, J.; Cowan, P.; Dionne, R.; Galer, B. S.; Hertz, S.; Jadad, A. R.; Kramer, L. D.; Manning, D. C.; Martin, S.; McCormick, C. G.; McDermott, M. P.; McGrath, P.; Quessy, S.; Rappaport, B. A.; Robbins, W.; Robinson, J. P.; Rothman, M.; Royal, M. A.; Simon, L.; Stauffer, J. W.; Stein, W.; Tollett, J.; Wernicke, J.; Witter, J. Core Outcome Measures for Chronic Pain Clinical Trials: IMMPACT Recommendations. Pain. 2005, pp 9–19. https://doi.org/10.1016/j.pain.2004.09.012. [19] Williams, D. A.; Arnold, L. M. Measures Applied to the Assessment of Fibromyalgia. Arthritis Care Res. 2011, 63 (2), 1487–1495. https://doi.org/10.1002/acr.20531.Measures.
We also realised that we described the outcomes of ‘pain’ and ‘function’ whereas the BPI does not strictly measure function but ‘pain interference’. Therefore, we have changed ‘function’ to ‘interference’ throughout the text.
|
|
Comments 6: First, in the results section you could provide a short introductory paragraph to help the reader. In addition, you could restructure the way in which the results are presented (e.g., by type of analysis) and their order (you offer satisfaction results that you only measure at the end of the program before those of the change between pre and post). |
|
Response 6: It is unusual in our experience to provide an introduction to the Results section. However, we did provide an introduction (summary of results) at the start of the Discussion section to help the reader.
Line 292 This study aimed to evaluate the feasibility and effectiveness of a new, 8-week (15 minutes per day) somatosensory perceptual training programme performed using a smartphone and vibrotactile pods in people with FMS. The study showed that the programme could be performed entirely remotely at home, with no face-to-face contact and the equipment sent by post. Participants adhered well to the programme and found the game enjoyable and easy to use. Positive effects were found on pain and interference with function scores, with effect sizes >0.7 for average pain intensity, highest pain intensity in the last 24 hours, and all of the function-related items of the BPI. Furthermore, 38% of participants reported a strong or very strong improvement in their condition at the end of the programme.
We structured the results in the order they were used as outcomes, even if this order was not chronological in terms of assessment time points. Our primary aim was to evaluate feasibility; therefore, we present feasibility data first (adherence [added following suggestions by reviewer 1] and satisfaction). The secondary aim was to evaluate effectiveness, therefore pain and function outcomes (BPI and perceived overall change) were presented after. We have changed the order of the discussion to follow the same order.
|
|
Comments 7: The discussion section might be improved, first, remembering what the objective of their work is. |
|
Response 7: Thank you for this suggestion. We have now restated the aim at the start of the discussion.
Line 292: This study aimed to evaluate the feasibility and effectiveness of a new, 8-week (15 minutes per day) somatosensory perceptual training programme performed using a smartphone and vibrotactile pods in people with FMS. |
|
Comments 8: The conclusions section might be improved, for example offering in more detail what are the potential practical implications of the results obtained. |
|
Response 8: We thank you for this suggestion. We did not feel it was appropriate to add such detail in the Conclusion, however we have added it in the discussion. Although the results of this study, which will need to be confirmed in a randomised controlled trial with a larger sample, suggest a real clinical effect of this training, they do not imply that this approach is beneficial for all individuals with FMS or that it can be used as a standalone treatment. Firstly, the management of FMS must be multidisciplinary [25],[26]. FMS is a chronic pain syndrome that is often associated with psychosocial factors that can contribute to the persistence of pain [27] and physical deconditioning. Somatosensory perceptual training should ideally be combined with other non-invasive approaches (selected according to the individual’s needs and preferences) that have also shown some efficacy, such as pain neurophysiology education [28], graded physical activities [29],[30], relaxation techniques [31], and cognitive-behavioural therapy [32] (to manage potential stress, anxiety, depression, etc.). A multimodal, and even multidisciplinary treatment approach combining individualised care with physical and psychosocial interventions, is therefore recommended. The TrainPain system could be provided to individuals with FMS as part of such a programme, with the aim of contributing to reducing pain symptoms. Remote use of the app could help individuals take charge of at least a part of their own rehabilitation and improve their level of self-efficacy.
|
|
Comments 9: Please check the references both in the text and references section, it does not seem to completely follow the format indicated by the journal. Likewise, if possible, it could be interesting to incorporate a greater number of references, especially more recent ones from 2024. |
|
Response 9: We corrected the format of the references and added 6 new references. However, we were unable to find references about somatosensory training for fibromyalgia in 2024.
[10] Barker, K. L.; Elliott, C. J.; Sackley, C. M.; Fairbank, J. C. T. Treatment of Chronic Back Pain by Sensory Discrimination Training. A Phase I RCT of a Novel Device (FairMed) vs. TENS. BMC Musculoskelet. Disord. 2008, 9, 1–8. https://doi.org/10.1186/1471-2474-9-97. [11] Kälin, S.; Rausch-Osthoff, A. K.; Bauer, C. M. What Is the Effect of Sensory Discrimination Training on Chronic Low Back Pain? A Systematic Review. BMC Musculoskelet. Disord. 2016, 17 (1), 1–9. https://doi.org/10.1186/s12891-016-0997-8. [15] Wang, F.; Ruberg, S. J.; Gaynor, P. J.; Heinloth, A. N.; Arnold, L. M. Early Improvement in Pain Predicts Pain Response at Endpoint in Patients with Fibromyalgia. J. Pain 2011, 12 (10), 1088–1094. https://doi.org/10.1016/j.jpain.2011.05.002. [16] Dworkin, R. H.; Turk, D. C.; Farrar, J. T.; Haythornthwaite, J. A.; Jensen, M. P.; Katz, N. P.; Kerns, R. D.; Stucki, G.; Allen, R. R.; Bellamy, N.; Carr, D. B.; Chandler, J.; Cowan, P.; Dionne, R.; Galer, B. S.; Hertz, S.; Jadad, A. R.; Kramer, L. D.; Manning, D. C.; Martin, S.; McCormick, C. G.; McDermott, M. P.; McGrath, P.; Quessy, S.; Rappaport, B. A.; Robbins, W.; Robinson, J. P.; Rothman, M.; Royal, M. A.; Simon, L.; Stauffer, J. W.; Stein, W.; Tollett, J.; Wernicke, J.; Witter, J. Core Outcome Measures for Chronic Pain Clinical Trials: IMMPACT Recommendations. Pain. 2005, pp 9–19. https://doi.org/10.1016/j.pain.2004.09.012.
[19] Williams, D. A.; Arnold, L. M. Measures Applied to the Assessment of Fibromyalgia. Arthritis Care Res. 2011, 63 (2), 1487–1495. https://doi.org/10.1002/acr.20531.Measures. [23] Sanz-Baños, Y.; Pastor-Mira, M. Á.; Lledó, A.; López-Roig, S.; Peñacoba, C.; Sánchez-Meca, J. Do Women with Fibromyalgia Adhere to Walking for Exercise Programs to Improve Their Health? Systematic Review and Meta-Analysis. Disabil. Rehabil. 2018, 40 (21), 2475–2487. https://doi.org/10.1080/09638288.2017.1347722.
|
Round 2
Reviewer 1 Report
Comments and Suggestions for Authors
I appreciate the response of the authors to my comments. Importantly, they have provided much-needed justification and background information on the brief pain index used in the study. They have also stated their hypotheses, and clearly defined what is their metrics are for measuring "feasibility".
As the only reviewer of the article, I feel it is my duty to be more stringent in the demands of the authors so that the quality of their work is high. Therefore I would like to see this minor revision:
1) In your discussion section please reiterate your hypotheses, and in the appropriate paragraphs, add a sentence explaining/justifying whether your results confirm or reject each one.
2) At line 300, you report that adherence at 75% was high and that [23] showed moderate to high attendance. I looked up this article. It states that adherence was 73% to 87.20%. This means that 73% was moderate and that high is 87.20%. Why not just list the range and let the readers decide for themselves? You can just state that is in the nominal range of the cited paper.
3) At Line 304 it is stated that the high adherence can be explained by several factors. Now that I think about it this is a rather strong statement to make, I believe [23]. I would like to see this statement toned down in its certainty. For example, saying that the result could "potentially be explained by".
Author Response
Thank you very much for your very positive feedback. We are very pleased that the changes we made to our article have met your expectations.
Comments 1:
In your discussion section please reiterate your hypotheses, and in the appropriate paragraphs, add a sentence explaining/justifying whether your results confirm or reject each one.
Response 1:
As suggested we refered to our hypotheses in the discussion section:
"In accordance with our hypotheses that participants would adhere to the programme, and that it would be associated with a good level of satisfaction, this study demonstrated the feasibility of an entirely remote, 2-month programme (vs. 4 weeks with a face-to-face programme initiation in the previous pilot study [13]) and that it was appreciated. With 75% of sessions performed by our participants, their adherence was in the range of the moderate to high attendance (73% to 87.20%) of supervised exercise sessions reported in people with FMS in a literature review [23]. This result is very encouraging for an entirely remote exercise programme. The high rate found in our study corresponds with the high level of satisfaction with the programme (median 9/10). These results could potentially be explained by the fact the participants found the training enjoyable and easy to complete, and that 80% perceived an overall positive change in their condition. Furthermore, 94% stated they would recommend the programme to others with similar pain. "
"With regards to the clinical effects, we hypothesised that pain intensity and interference would be reduced after completion of the programme and that participants would perceive an overall positive change in their condition. In line with our expectations, our study showed significant improvements in most pain and interference scores "
Comment 2: At line 300, you report that adherence at 75% was high and that [23] showed moderate to high attendance. I looked up this article. It states that adherence was 73% to 87.20%. This means that 73% was moderate and that high is 87.20%. Why not just list the range and let the readers decide for themselves? You can just state that is in the nominal range of the cited paper.
Response 2: Thank you for pointing this out. As suggested we have changed the sentence into:
"With 75% of sessions performed by our participants, their adherence was in the range of the moderate to high attendance (73% to 87.20%) of supervised exercise sessions reported in people with FMS in a literature review [23]. "
Comment 3: At Line 304 it is stated that the high adherence can be explained by several factors. Now that I think about it this is a rather strong statement to make, I believe [23]. I would like to see this statement toned down in its certainty. For example, saying that the result could "potentially be explained by".
Response 3: thank you for your comment. You are right. We have changed our sentence and stated that "the result could potentially be explained by"."
Reviewer 2 Report
Comments and Suggestions for Authors
Thank you very much for giving me the opportunity to review again this interesting paper entitled Feasibility of an 8-week Home-Based Sensory Perception Training Game for People with Fibromyalgia: A Pilot Study. Altogether it is an interesting and well written study that aimed to investigate the feasibility of an 8-week home-based somatosensory, entirely remote, self-training programme to reduce pain in people with fibromyalgia syndrome (FMS) as well as to perform a preliminary evaluation of its effect on pain and function among 29 individuals (of a total of 34 individuals recruited) with FMS.
The authors have improved the manuscript, and I would just like to point out that it would be interesting to carefully consider the formatting issues, for example there are too many spaces between paragraphs.
Author Response
Comments: The authors have improved the manuscript, and I would just like to point out that it would be interesting to carefully consider the formatting issues, for example there are too many spaces between paragraphs.
Reply: thank you very much for your very positive feedback. We are very pleased that the changes we made to our article have met your expectations. Regarding your comment about the formatting issues, we have deleted some spaces between paragraphs.